# Prospects for the Preservation of the Main *Pinus sylvestris* L. Ecotypes in Poland in the Context of the Habitat Conditions of Their Occurrence

Monika Konatowska [1,*] , Adam Młynarczyk [2] and Paweł Rutkowski [1]

[1] Department of Botany and Forest Habitats, Faculty of Forestry and Wood Technology, Poznań University of Life Sciences, Wojska Polskiego 71F, 60-625 Poznań, Poland; redebede@wp.pl
[2] Environmental Remote Sensing and Soil Science Research Unit, Faculty of Geographic and Geological Sciences, Adam Mickiewicz University in Poznań, Wieniawskiego 1, 61-712 Poznań, Poland; adam.mlynarczyk@amu.edu.pl
* Correspondence: monika.konatowska@up.poznan.pl

**Abstract:** This study investigates the prospects for preserving the main *Pinus sylvestris* L. (Scots pine) ecotypes in Poland, considering the habitat conditions of their occurrence. Scots pine is known for its wide distribution and natural adaptability to various habitats. However, there is an increasing vulnerability of pine forests to damage from biotic factors and a decrease in natural regeneration, particularly in areas under legal protection. Additionally, projected climate change has raised concerns about the future of *Pinus sylvestris*, placing it in the "losing" group of tree species. The aim of the study was to analyze the habitat conditions of the seven main selected *Pinus sylvestris* L. ecotypes to assess the sustainability of pine stands in their natural habitat conditions. Out of the seven populations of studied pine ecotypes, only one grows under conditions representing a typical form of pine forest (*Leucobryo–Pinetum* plant association). Two populations grow under conditions corresponding to potential deciduous forests (*Galio sylvatici–Carpinetum* and *Calamagrostio arundinaceae–Quercetum petraeae*). The remaining populations represent potentially mixed oak–pine forests. Such a distribution of plant communities, except for *Leucobryo–Pinetum*, does not guarantee the continuity of the studied pine stands as a result of their natural regeneration. Therefore, it is necessary to preserve the offspring of the studied populations outside their occurrence sites, but the studied pine stands should be preserved until their natural death in their natural habitats. In the conducted research, the NDVI turned out to be very useful, showing a high correlation with the trophicity of habitat expressed in the diversity of plant communities, as well as with the height and diameter of the studied stands.

**Keywords:** mast pines; soils; plant association; NDVI





## 1. Introduction

*Pinus sylvestris* L. (Scots pine) is the most widely distributed pine species in the world [1]. Due to its natural plasticity, it colonizes a wide range of habitats [2]. However, increasing susceptibility of pine to damage caused by biotic factors, such as mistletoe [3], and a decrease in the natural regeneration of pine forests in areas under legal protection has been observed [4]. Dyderski et al. [5], considering the effects of climate changes, classified *Pinus sylvestris* as a "loser" among other studied tree species. Not only today but also in the past, Scots pine has been a very important species in several European countries due to its valuable wood. During the 18th and 19th centuries, the best mast wood was imported from Riga (present-day Latvia), and the desirable "Riga pine" wood reached very high prices. The term "Riga pine" referred more to a pine that met certain technical criteria than to a pine growing in the vicinity of Riga or imported through that port [6]. One of the ecotypes of pine growing in Poland, where the share of pine in the country's forest area is 58.2% [2], met the criteria for "Riga pine" and was called "Supraska pine" or "Mast pine" in the

past [7]. In the work of Daszkiewicz and Oleksyn] [6], it is worth emphasizing the words cited from Duhamel in 1767 regarding "Riga pine", which drew attention to the significant influence of tree age, the quality of the habitat where it grew, and the climate on the quality of mast. Therefore, one could ask whether "Riga pine" was an exceptional ecotype or whether the exceptional conditions were the habitat conditions in which this pine grew. Nowadays, the question of habitat conditions is primarily important for the future of the most valuable pine ecotypes. Temperature, humidity, light, and the interactions between these factors [8,9] are among the most important factors influencing the natural regeneration of pine trees, but competition from other plants is equally important [10]. Therefore, the aim of our study was to analyze the habitat conditions of selected *Pinus sylvestris* L. ecotypes in terms of the possibility of ensuring the sustainability of pine stands in their natural conditions. The following research hypothesis was put forward: the current habitat conditions and directions of vegetation development do not guarantee the preservation of valuable *Pinus sylvestris* ecotypes in places of their natural occurrence. We adopted the term "ecotype" following the literature on the studied stands [6,7,11,12], corresponding to the aim of the study and also to the classic definition of ecotype given by Krebs [13]. According to Krebs, an ecotype is "a group of populations of one species that are adapted to specific climatic and habitat conditions". As the climate is an element of the habitat, separating both terms in the above definition may be considered an error; however, the essence of the definition corresponds to the concept of our studies.

Additionally, to assess the health status of the studied pine ecotypes, the Normalized Difference Vegetation Index (NDVI) was taken into account as a parameter of their current condition. NDVI, as an indicator of *Pinus sylvestris* health, can be influenced by several factors, including climate, habitat conditions, and biotic interactions. Therefore, we also assumed that the use of NDVI could be treated as a result of habitat conditions affecting the future of the studied Scots pine ecotypes.

## 2. Materials and Methods

### 2.1. Research Area

Seven study plots were selected for the study. Their location and basic data are presented in Figure 1 and Table 1. "Taborska" (Table 1, number 1) and "Supraska" (4) pine ecotypes were selected based on the results of Jelonek et al. [7] and Barzdajn [14]. The ecotype from Kampinoski National Park (2 and 3) was selected based on the results of Przybylski et al. [2]. The National Park Bory Tucholskie (6) was chosen due to the results of Młynarczyk et al. [15]. The "Rychtalska pine" ecotype (5) was chosen due to the results of Wójkiewicz et al. [16] and Barzdajn [14]. The ecotype from Notecka Forest (7) was chosen as the oldest pine stand in one of the biggest pine complexes in Poland.

Geometric data of research plots, available in the SHP format, together with the database in the following formats: .prj, .sbn, .sbx, and .shx. They were obtained from the Forest Data Bank—a database where the results of habitat research conducted on State Forests' lands by specialized expert teams are collected (https://www.bdl.lasy.gov.pl/portal/; accessed on 11 December 2022). The database is publicly available in two language versions—Polish as the primary language and English.

From the same data repository, the description of research plots was obtained, including the age of pine stands, type of soil, and potential plant community classified according to Matuszkiewicz [17].

Descriptions of the research areas were supplemented with a summary of atmospheric precipitation according to Muñoz Sabater [18].

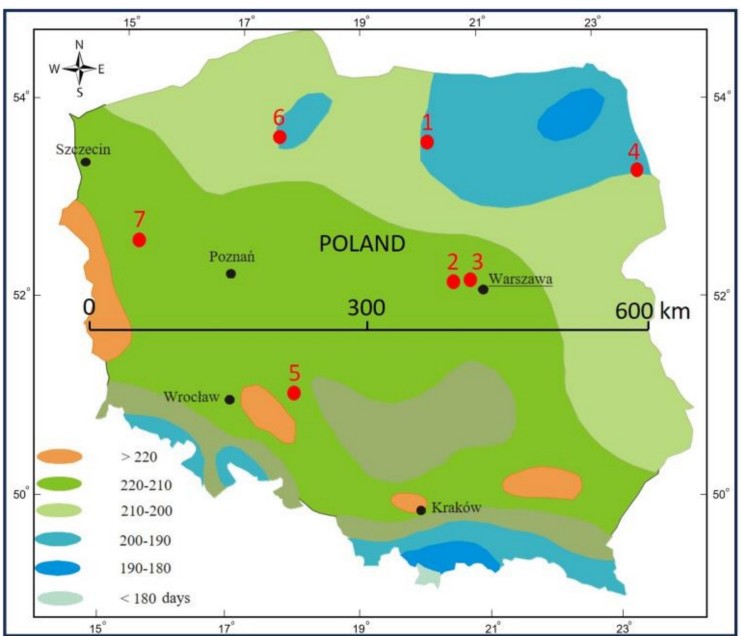

**Figure 1.** Location of studied pine ecotypes (red points) on the basis of a map depicting the length of the vegetation period in Poland: 1—Taborska Pine; 2—Kampinoski NP-01; 3—Kampinoski NP-02; 4—Supraska Pine; 5—Rychtalska Pine; 6—Bory Tucholskie NP; 7—Notecka Pine.

**Table 1.** Description of studied pine stands.

| No | Name | Coordinates of the Middle of Research Plot | Age (2022) | Area (ha) |
|----|------|--------------------------------------------|------------|-----------|
| 1 | Taborska Pine | 53°46′30.8″ N; 20°01′32.1″ E | 268 | 15.20 |
| 2 | Kampinoski NP-01 | 52°20′26.3″ N; 20°23′41.8″ E | 188 | 19.05 |
| 3 | Kampinoski NP-02 | 52°20′12.3″ N; 20°46′33.3″ E | 183 | 12.59 |
| 4 | Supraska Pine | 53°15′08.3″ N; 23°29′44.9″ E | 196 | 3.14 |
| 5 | Rychtalska Pine | 51°11′18.4″ N; 17°56′55.3″ E | 142 | 15.15 |
| 6 | Bory Tucholskie NP | 53°47′44.3″ N; 17°34′46.1″ E | 140 | 19.28 |
| 7 | Notecka Pine | 52°42′08.0″ N; 15°28′15.1″ E | 133 | 3.03 |

*2.2. Calculation of the NDVI*

Młynarczyk et al. [15], after testing 249 vegetation indices calculated according to the formulas stored and described in the "Index Database" (https://www.indexdatabase.de/, accessed on 28 December 2022), demonstrated that the NDVI can be used as an indicator describing both the moisture and trophic status of forest habitats. This was considered an important measure of habitat conditions, which simultaneously influences the condition of forest stands. Therefore, NDVI data were calculated and used according to the methodology described by Młynarczyk et al. [15]. Following the authors' recommendations, the NDVI was calculated for three months: July, August, and September (2020), which best reflect the NDVI for the temperate climate zone where Poland is located.

Due to the age of the forest stands and their resulting thinning, which in most cases leads to vigorous development of the lower developmental layers of the forest stand and understory, the median rather than the mean NDVI value was taken into account when ranking the studied populations of pine trees. The median value could be influenced not only by areas covered by pine forest but also by other plant species.

*2.3. Statistical Analysis*

Since the most valuable forest stands in Poland were selected for the study, based on their age, quality, and representativeness for large pine forest complexes, the number of research areas is limited (7). Therefore, the statistical analysis was restricted to

a method based on ranks assigned to the data parameters (tree slenderness coefficient, plant community, NDVI) and the calculation of Pearson's correlation coefficients between parameters.

## 3. Results

Soil descriptions, potential vegetation types for individual research areas, as well as the average diameter and height of forest stands in the research areas, are shown in Table 2. In Table 3 mean, median, and maximum NDVI values are presented.

**Table 2.** Soil type, potential plant community of research plots, average height (Av. H), average diameter at breast height of stands (Av. DBH), and Tree Slenderness Coefficient (H/D) of tested stands.

| No | Name | Soil Type | Potential Plant Community | Av. H (m) | Av. DBH (cm) | (H/D) ×100 |
|----|------|-----------|---------------------------|-----------|--------------|------------|
| 1 | Taborska Pine | PRS * | Eutrophic oak–hornbeam forest (*Galio sylvatici–Carpinetum* plant community) | 33 | 66 | 50 |
| 2 | Kampinoski NP-01 | PS | Oligotrophic *Quercus robur–Pinus sylvestris* forest (*Querco roboris–Pinetum*) | 25 | 45 | 56 |
| 3 | Kampinoski NP-02 | PS | Oligotrophic *Quercus robur–Pinus sylvestris* forest (*Querco roboris–Pinetum*) | 24 | 46 | 52 |
| 4 | Supraska Pine | TRS | Mesotrophic *Quercus robur–Pinus sylvestris* forest (*Seratulo–Pinetum*) | 34 | 49 | 69 |
| 5 | Rychtalska Pine | BRS | Mesotrophic *Quercus petraea* forest (*Calamagrostio Arundinaceae–Quercetum petraeae*) | 30 | 41 | 73 |
| 6 | Bory Tucholskie NP | PS | Oligotrophic pine forest (*Leucobryo–Pinetum*) | 22 | 33 | 67 |
| 7 | Notecka Pine | PRS | Oligotrophic *Quercus robur–Pinus sylvestris* forest (*Querco roboris–Pinetum*) | 26 | 37 | 70 |

\* PRS—podzolic rusty soil; BRS—brown rusty soil; TRS—typical rusty soil; PS—podzolic soil.

**Table 3.** NDVI value in 2020.

| No | Name | Mean | | | | Median | | | | Standard Deviation | | |
|----|------|------|------|------|------|------|------|------|------|------|------|------|
| | | Jul. | Aug. | Spt. | Av. | Jul. | Aug. | Spt. | Av. | Jul. | Aug. | Spt. |
| 1 | Taborska Pine | 0.794 | 0.793 | 0.793 | 0.793 | 0.885 | 0.881 | 0.880 | 0.882 | 0.262 | 0.261 | 0.261 |
| 2 | Kampinoski NP-01 | 0.822 | 0.832 | 0.784 | 0.812 | 0.823 | 0.828 | 0.785 | 0.812 | 0.013 | 0.021 | 0.012 |
| 3 | Kampinoski NP-02 | 0.760 | 0.787 | 0.758 | 0.768 | 0.756 | 0.785 | 0.750 | 0.764 | 0.016 | 0.014 | 0.025 |
| 4 | Supraska Pine | 0.580 | 0.588 | 0.593 | 0.587 | 0.812 | 0.822 | 0.815 | 0.816 | 0.367 | 0.372 | 0.375 |
| 5 | Rychtalska Pine | 0.885 | 0.853 | 0.859 | 0.866 | 0.888 | 0.851 | 0.859 | 0.866 | 0.019 | 0.014 | 0.013 |
| 6 | Bory Tucholskie NP | 0.788 | 0.808 | 0.682 | 0.759 | 0.789 | 0.808 | 0.677 | 0.758 | 0.012 | 0.010 | 0.019 |
| 7 | Notecka Pine | 0.718 | 0.737 | 0.746 | 0.734 | 0.719 | 0.742 | 0.757 | 0.740 | 0.033 | 0.039 | 0.035 |

All the soils in which the studied stands grow are sandy. Table 2 adopts the Polish nomenclature for the names of soil types and subtypes used in the Polish Forest Soil Classification [19], which does not always fully correspond to the international soil classification system for naming soils [20]. According to the WRB characteristics, rusty soils are similar to Arenosols, although, in the Polish forest soil classification, Arenosols are a separate type of soil. A common feature of the soils in which the studied stands grow is their acidic reaction. Despite similar characteristics, the potential of the soils listed in Table 1 is different, which is expressed in the diversity of potential plant communities listed in the table, determined on the basis of a specific combination of forest undergrowth species. Of these, the *Leucobryo–Pinetum* association is considered to be the poorest, typically pine forest in Poland. The richest association is considered to be *Galio sylvatici–Carpinetum*. Taking into account the remaining mentioned plant communities, it can be assumed that the habitats of

the studied pine stands represent a full cross-section of trophic mesic habitat types, from typical pine forests to potential broadleaf forests, including hornbeam and oak.

Table 3 presents data from three months representing the full growing season (July–September), thus avoiding differences in NDVI values resulting from changes in the season. Taking into account the data from Table 2 indicating the similar nature of the soils (sandy, acidic soils), it can be assumed that the differences between the studied ecotypes, as well as within a given ecotype in individual months, may result from weather differences affecting the condition of the studied tree stands. The studied stands are located in various regions of Poland, with variable climatic conditions (Figure 1), which affects both the length of the growing season and other weather indicators. The studied tree stands grow on sandy soils with deep groundwater levels; thus, the distribution of atmospheric precipitation is an important factor influencing the condition of the trees. The periodic lack of rainfall in a given location may affect differences in the condition of tree stands in individual months, which could explain the NDVI values for the Bory Tucholskie ecotype between July and August (0.788 and 0.808, respectively) and September (0.682). However, the interpretation of the data may be difficult due to the fact that the local rainfall distribution may differ from the source data and also because tree stands may react with varying delays to rainfall or a lack thereof. Table 4 lists the monthly rainfall totals from June 2020 to September 2020. Table 3 shows the NDVI values from July to September, which is one month shorter. However, it was assumed that rainfall in June may affect the condition of tree stands in July; hence, June was added to Table 4.

**Table 4.** Precipitation (mm) from June 2020 to September 2020 compared to the average median value for NDVI for the studied pine stands.

| Ecotype Name | Jun. | Jul. | Aug. | Spt. | Av. | Sum | NDVI Median |
|---|---|---|---|---|---|---|---|
| Taborska Pine | 93 | 137 | 29 | 32 | 72.8 | 291 | 0.882 |
| Kampinoski NP-01 | 141 | 62 | 128 | 47 | 94.5 | 378 | 0.812 |
| Kampinoski NP-02 | 99 | 77 | 136 | 51 | 90.8 | 363 | 0.764 |
| Supraska Pine | 71 | 70 | 130 | 24 | 73.8 | 295 | 0.816 |
| Rychtalska Pine | 83 | 58 | 30 | 63 | 58.5 | 234 | 0.866 |
| Bory Tucholskie NP | 50 | 111 | 25 | 28 | 53.5 | 214 | 0.758 |
| Notecka Pine | 26 | 120 | 41 | 37 | 56 | 224 | 0.740 |

The rainfall sums given in Table 4 correspond to some extent to the NDVI values given in Table 3, indicating the lowest NDVI values for pine stands in the areas with the lowest rainfall totals (Bory Tucholskie and Notecka Pine); however, the Pearson correlation coefficient calculated for the data from the columns "sum" and "NDVI median" is only 0.10.

In order to determine the prospects for maintaining the natural continuity of the studied pine forest stands, the tree slenderness coefficient, plant community, and the NDVI were ranked and assigned appropriate ranks:

- Tree Slenderness Coefficient (TSC): 1—lowest slenderness coefficient, 7—highest slenderness coefficient; the lower the TSC, the greater the resistance to wind damage.
- Plant community: 1—*Leucobryo–Pinetum*, 2—*Querco roboris–Pinetum*, 3—*Seratulo–Pinetum*, 4—*Calamagrostio arundinaceae–Quercetum petraeae*, 5—*Galio sylvatici–Carpinetum*; the lower the index, the higher the likelihood of natural pine regeneration.
- NDVI: 1—lowest median value for the NDVI, 7—highest median value for the NDVI; the higher the NDVI value, the better the habitat conditions and, therefore, the lower the chance of natural pine regeneration due to competition from deciduous tree species.

Between the individual ranking categories shown in Table 5, correlation coefficients were calculated, resulting in the values presented in Table 6. Additionally, the Pearson correlation coefficient between the average height and average diameter at breast height of stands representing a given pine ecotype and NDVI is calculated and presented in Table 6.

**Table 5.** Ranking of studied pine forest stands based on analyzed characteristics.

| | Rank | | | |
|---|---|---|---|---|
| | TSC | Plant Community (Competition) | NDVI | Sum |
| Taborska Pine | 1 | 5 | 7 | 13 |
| Kampinoski NP-01 | 3 | 2 | 4 | 9 |
| Kampinoski NP-02 | 2 | 2 | 3 | 7 |
| Supraska Pine | 5 | 3 | 5 | 13 |
| Rychtalska Pine | 7 | 4 | 6 | 17 |
| Bory Tucholskie NP | 4 | 1 | 2 | 7 |
| Notecka Pine | 6 | 2 | 1 | 9 |

**Table 6.** Correlation coefficients between tree slenderness coefficient, plant community, and the NDVI calculated on the basis of ranks adopted in Table 5.

| Parameter | Correlation Coefficients |
|---|---|
| TSC/Plant community | –0.11 |
| TSC/NDVI | –0.21 |
| H/NDVI | 0.74 |
| D/NDVI | 0.70 |
| Plant community/NDVI | 0.89 |

The high correlation coefficient between the NDVI and plant community and also NDVI and height and average diameter at breast height of stands representing a given pine ecotype confirms the role of the NDVI as a synthetic indicator of habitat trophicity.

A summary of the average height of the studied pine stands, potential plant communities in their locations, soil characteristics of the habitat for each ecotype, and the NDVI is presented in Figure 2.

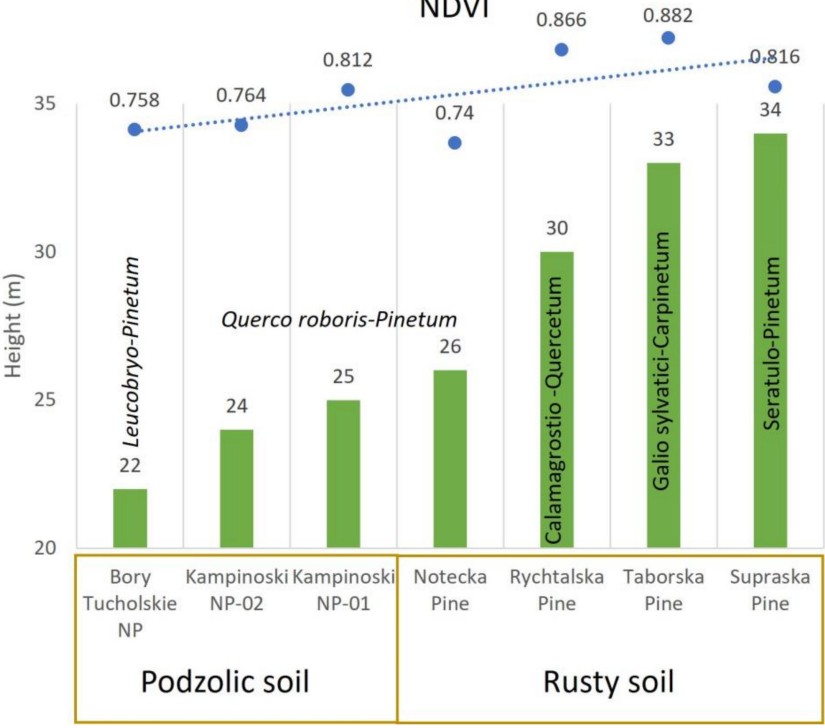

**Figure 2.** Summary of research results for seven ecotypes of *Pinus sylvestris*, including soil type, plant community, NDVI (median), and average tree stand height.

As shown by the collected data (Figure 2), the assessment of each population varies depending on the adopted criterion. The ranking of all features presented in Table 5, which reflects the perspective of maintaining the natural continuity of the studied pine stands, allows the following order for the tested populations: Bory Tucholskie NP and Kampinoski NP-02 (highest probability of stand continuity; lowest sum of ranks = 7) → Kampinoski NP-01 and Notecka Forest (sum of ranks = 9) → Taborska Pine and Supraska Pine (sum of ranks = 13) → Rychtalska Pine.

## 4. Discussion

Maintaining forest continuity is a fundamental obligation in forestry management. However, it is important to distinguish between maintaining forest continuity in any form and with a specific species composition. As demonstrated by Przybylski et al. [2], in the case of Scots pine, the dominant tree species in Poland, the ease of establishing pine stands through artificial regeneration leads to less attention being paid to factors limiting its natural regeneration. At the same time, a deteriorating condition of pine stands established through artificial regeneration is observed [2]. Therefore, the protection of key pine ecotypes in their natural habitats is considered an important aspect in the discussion on forestry management and the conservation of valuable natural resources represented by pine stands. In the presented research results, the study focused on the soil conditions of the examined stands, the potential plant communities, NDVI, and the diameter and height of trees, which are correlated with the tree slenderness coefficient. All these factors are interconnected with the response of plants (trees) to habitat conditions, but they are also influenced by forestry practices. As shown in Table 2, all the studied stands are located on podzolic or rusty soils, which are considered to be some of the poorest soil types in Poland due to their sandy granulometric composition. As a result, these soils were afforested with pine, which is considered a pioneer species [1]. Since rusty soils cover half of Poland's forest soils, taking into account the share of podzolic soils, the dominant role of Scots pine is highlighted, and a share of 60% in Poland is obtained [2].

As indicated in Table 2, the potential plant community of the studied pine stands includes not only poor pine forest communities but also potential deciduous forest plant associations. Among the studied stands, the only pine-specific plant community is *Leucobryo–Pinetum* [17], although, in Germany, some of the stands sampled as representative of the *Leucobryo–Pinetum* association were probably located on sites formerly occupied by pine–oak or birch–oak forests [21]. The other plant communities listed in Table 2 potentially have a smaller or larger proportion of deciduous species, with the oak–hornbeam forest (*Galio sylvatici–Carpinetum*) and mesotrophic sessile oak forest (*Calamagrostio arundinaceae–Quercetum petraeae*) representing deciduous forest stands.

It should be noted that the age of the studied stands exceeds 180 years in most cases (up to a maximum of 268), which means that these stands were established 200 years ago under cool climatic conditions, which favored coniferous species. Currently, the warming climate favors deciduous species [5]. Consequently, it is not excluded that the development of vegetation in the studied pine stands reflects the consequences of climate change, resulting in the transformation of former pine forest habitats into deciduous forests. As a consequence, competition from deciduous species hinders natural pine regeneration by shading the forest floor. It is worth noting that the observation of these changes is possible due to the exclusion of the studied stands from forest management and the attainment of advanced age by the pine trees. Under typical forestry management in Poland, pine would be harvested after reaching the age of 100–110 years, and pine would generally be reintroduced on sandy soils, as described by Konatowska and Rutkowski [22]. The situation described by both authors is illustrated in Figure 3.

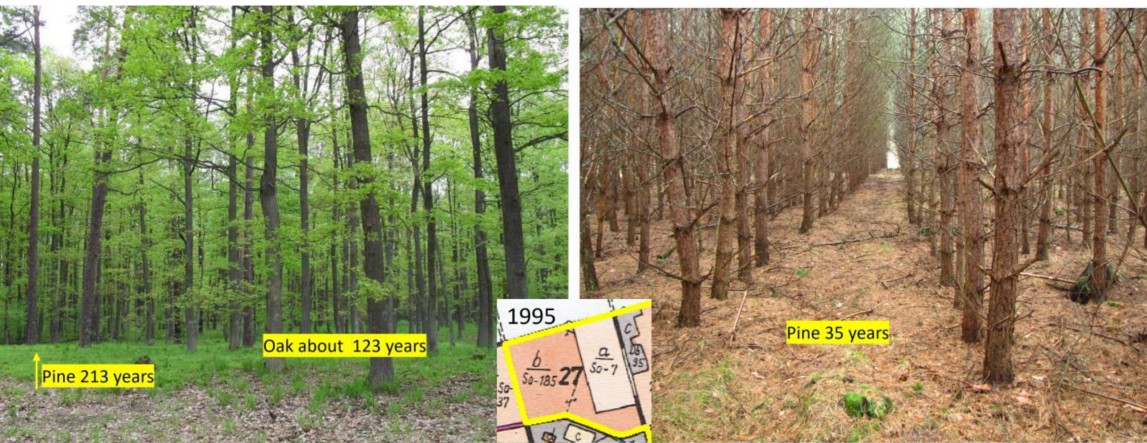

**Figure 3.** Comparison of a 213-year-old pine stand with a developed oak stand underneath (**left**) and a pine monoculture planted in place of a cut oak stand (**right**); in the middle, a map of the stands from 1995, showing two forest compartments: "a"—with 7-year-old pine (So) monoculture (actually 35) and "b"—with 185-year-old pine (So) (actually with single specimens of pine 213 and domination of oak about 123 years old).

As shown in Figure 3, leaving the pine stand to natural decay allows for successive changes leading from pine forests to deciduous forest communities. However, in such cases, the longevity of valuable pine ecotypes ends with the death of the oldest pine specimens. This does not necessarily mean the irreversible loss of a particular ecotype if genetic material was previously collected, but planting offspring of valuable pine specimens in different habitat conditions does not guarantee similar growth parameters, as noted by Daszkiewicz and Oleksyn [6] when discussing the Riga pine and posing the question of whether it was an exceptional ecotype or the habitat conditions in which the pine grew were exceptional. Barzdajn's research results [14] indicate that the Taborska, Supraska, and Rychtalska pines stand out in terms of average diameter at breast height and height compared to other European populations, but the results may have been influenced by habitat conditions less favorable for populations from the southern part of Europe, as pointed out by the author himself (the research was conducted within the range of the Supraska ecotype). On the other hand, research by Szeligowski et al. [23] showed that out of the 16 tested Polish populations, Rychtalska, Taborska, and Supraska pines ranked 7th, 12th, and 14th, respectively, in terms of average diameter at breast height, and Rychtalska pine ranked 3rd, Taborska pine ranked 8th, and Supraska pine ranked 14th in terms of height. Among the three mentioned pines, Rychtalska pine performed best in both parameters, followed by Taborska pine and Supraska pine. In the described studies, the Bolewice ecotype performed the best in terms of diameter at breast height and tree height. Conversely, in research by Remlein et al. [24], Bolewice pines were characterized by the lowest average values of the described morphological characteristics. Regardless of the divergent results, the populations of studied pine stands in this research deserve protection due to their age. It can be assumed that their tree slenderness coefficient contributes to the longevity of the studied populations. According to the classification presented by Ige and Komolafe [21], trees with a TSC > 99 exhibit a high slenderness coefficient (prone to wind throw), those with a TSC of 70–99 show a moderate slenderness coefficient (able to withstand wind throw), and those with TSC < 70 have a low slenderness coefficient (also able to withstand wind throw). Our studied pines show TSC values between 52 and 73 (Table 2), indicating their relative resistance to wind activity, which may contribute to their age. The wind resistance of the 268-year-old Taborska pine stand, or the nearly 200-year-old Supraska pine stand may also confirm their classification as Mast pines in the past.

The validity of adopting TSC as a measure of the durability of a pine stand may also be confirmed by the results of research by Jarmuł and Kaczmarski [25], who showed that the slenderness increases with deterioration of the biosocial position of the tree. There-

fore, although the differences in levels between the tree slenderness coefficient and plant community rankings may not be equivalent, it seems that the ranking adopted for the slenderness coefficient (Table 5) was arranged correctly. However, the fact that the correlation between NDVI and TSC shown in Table 6 is statistically insignificant may be because most of the studied stands show similar TSC values, falling into one group distinguished by Ige and Komolafe [26]. As a side effect of the research, a correlation between the NDVI and site trophism expressed by plant communities (Table 6) has been demonstrated. The Normalized Difference Vegetation Index (NDVI), a widely used method for estimating vegetation greenness, and other spectral indices are commonly used to assess the forest environment [27–30], but studies on this index show the complex relationship between NDVI and ecological factors such as meteorological data, soil moisture, and vegetation cover type. Creating a trophic grid of forest habitats based on NDVI could be a subject of separate research. In the context of the conducted research, the Supraska Pine population deserves particular attention, as it achieves the highest average height but is associated with the *Seratulo–Pinetum* plant community, described as a subboreal mixed forest [14]. In Polish forest nomenclature, mixed forests mainly consist of conifers and are considered to be trophically less rich than oak or oak–hornbeam forests. The analysis of the NDVI index presented in Table 3 and Figure 2 would confirm this opinion. However, as Matuszkiewicz states [17], in terms of habitat, the *Seratulo–Pinetum* refers to thermophilous oak forests, considered the floristically richest forest community in Poland. Matuszkiewicz also adds that it seems that in the northeastern part of Poland, outside the natural range of thermophilous oak forests, the *Seratulo–Pinetum* community replaces them in analogous habitats. The harsh climate of this part of the country (Figure 1) can be considered the reason for the natural replacement of fertile oak forests by pine forests. In this context, the lower NDVI value compared to the described *Calamagrostio arundinacea–Quercetum* (Rychtalska Pine) and *Galio sylvatici–Carpinetum* (Taborska Pine) seems justified, once again demonstrating that NDVI can be a sensitive indicator of habitat trophism, reflecting not only soil conditions but also significant climatic conditions for vegetation. In this case, it can be assumed that the highest average height achieved by the Supraska Pine stands reflects soil fertility, but harsher climatic conditions limit the development of deciduous stands. The combination of these factors may have contributed to the formation of the ecotype described as Riga pine, also called "Mast pine" in the past.

## 5. Conclusions

Out of the seven studied pine ecotypes, only one grows under conditions representing a typical form of pine forests (*Leucobryo–Pinetum* plant association). Two populations grow under conditions corresponding to potential deciduous forests (*Galio sylvatici–Carpinetum* and *Calamagrostio arundinaceae–Quercetum petraeae*). The remaining populations represent potential mixed oak–pine forests. Such a distribution of plant communities, except for *Leucobryo–Pinetum*, does not guarantee the continuity of the studied pine stands as a result of their natural regeneration. Therefore, it is necessary to preserve the offspring of the studied populations outside their occurrence sites, but the studied pine stands should be preserved until their natural death in their natural habitats.

Supraska Pine deserves special attention due to its highest growth among all the studied ecotypes, growing under soil conditions corresponding to thermophilous oak forests but limited by harsher climatic conditions. The combination of these factors may have contributed to the formation of the ecotype described as Riga pine, also called "Mast pine" in the past.

As an important side effect, this research also demonstrates the correlation between the NDVI and habitat trophism expressed by the diversity of plant communities. For habitats potentially characterized by *Galio sylvatici–Carpinetum* vegetation (oak–hornbeam forest), the median NDVI for the full growing season was 0.882, while for the *Leucobryo–Pinetum* habitat (poor pine forest) it was 0.758.

**Author Contributions:** Methodology, M.K. and A.M.; Software, A.M.; Formal analysis, A.M.; Data curation, A.M.; Writing—original draft, M.K.; Writing—review & editing, P.R.; Visualization, M.K.; Supervision, P.R. All authors have read and agreed to the published version of the manuscript.

**Funding:** This research received no external funding.

**Data Availability Statement:** The data presented in this study are not publicly available, but may be obtained from the authors upon reasonable request.

**Conflicts of Interest:** The authors declare no conflict of interest.

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
