# Peer review of "Prospects for the Preservation of the Main Pinus sylvestris L. Ecotypes in Poland in the Context of the Habitat Conditions of Their Occurrence"

_forests, doi:10.3390/f14101967_

Round 1
Reviewer 1 Report
Dear authors,
The study s to analyze the habitat conditions of selected Pinus sylvestris L. ecotypes in terms of the possibility of ensuring the sustainability of pine stands in their natural condition. I congratulate for the study. In my opinion, it brings important knowledge and a great methodology, but it needs improvement in the results section.
Title: Should be in italic: Pinus sylvestris
Keywords: Must be different from the words in the title
Figure 1: North/Scale and coordinates are missing
Table 1: I suggest a new title, for example: Description of studied pine stands.
Results: This topic needs attention. No results are presented, just in tables. The authors must write about results. Of course, the information in the tables is important, but what can I see in the table? The author must give a direction for the readers. For example, the authors can explore the amplitude of NDVI for each stand. Has any difference between stands? The highest value was the same among stands? See Bory Tucholskie, NDVI incresead in aug and decreased in September. Was the same for all stands?
About organization: first you write/show the results, than you insert the table/figure that you cited in the beginning. For example, the text from Line 144-149 must be before Figure 2.
L111: Has a little confuse. It seems that started the name of table but also is talking about table 2 and 3.
Discussion:
Is the climate the same for all stands? Certainly not. Could you insert information about weather/climate for each stand in the Materials section?
Despite the necessity to improve results section, the discussion is great. I suggest to include something about climate change and it´s impact on the Pinus. The authors started this discussion in the introduction section. In my opinion its a good way return and finish the discussion with some appointment.
Author Response
Response to the Reviewers to the manuscript forests-2599299: “Prospects for the Preservation of the Main Pinus sylvestris L. Ecotypes in Poland in the Context of the Habitat Conditions of their Occurrence”.
We would like to thank all reviewers for their insightful and substantively valuable comments that allowed us to make our manuscript better. At the same time, we attach below our responses to the Reviewers' comments.
Reviewer 1 (R1).
Dear authors,
The study s to analyze the habitat conditions of selected Pinus sylvestris L. ecotypes in terms of the possibility of ensuring the sustainability of pine stands in their natural condition. I congratulate for the study. In my opinion, it brings important knowledge and a great methodology, but it needs improvement in the results section.
R1: Title: Should be in italic: Pinus sylvestris
Authors: The change has been made
R1: Keywords: Must be different from the words in the title
Authors: We removed “habitat conditions” and introduced soils.. The remaining keywords are not repeated
R1: Figure 1: North/Scale and coordinates are missing
Authors: The figure has been improved
R1: Table 1: I suggest a new title, for example: Description of studied pine stands.
Authors: We have changed the title of the table according to the reviewer's suggestion
R1: Results: This topic needs attention. No results are presented, just in tables. The authors must write about results. Of course, the information in the tables is important, but what can I see in the table? The author must give a direction for the readers. For example, the authors can explore the amplitude of NDVI for each stand. Has any difference between stands? The highest value was the same among stands? See Bory Tucholskie, NDVI incresead in aug and decreased in September. Was the same for all stands?
Authors: We have completed the comments to the tables
R1: About organization: first you write/show the results, than you insert the table/figure that you cited in the beginning. For example, the text from Line 144-149 must be before Figure 2.
Authors: The text has been reorganized
R1: L111: Has a little confuse. It seems that started the name of table but also is talking about table 2 and 3.
Authors: This is a bug we fixed. We thank the Reviewer for spotting it.
R1: Discussion:
R1: Is the climate the same for all stands? Certainly not. Could you insert information about weather/climate for each stand in the Materials section?
Authors: A table with rainfall totals has been added to the Results section to compare the impact of rainfall on variations in NDVI values
R1: Despite the necessity to improve results section, the discussion is great. I suggest to include something about climate change and it´s impact on the Pinus. The authors started this discussion in the introduction section. In my opinion its a good way return and finish the discussion with some appointment.
Reviewer 2 Report
The manuscript entitled “Prospects for the Preservation of the Main Pinus sylvestris L. Ecotypes in Poland in the Context of the Habitat Conditions of their Occurrence” describes a new approach to assess the sustainability of Pinus sylvestris L. stands in their natural habitat conditions through the analysis of 7 major selected ecotypes and their corresponding environmental conditions. This work could provide valuable information for recognizing the developmental prospects of major Pinus sylvestris L. ecotypes in Poland. I recommend to a minor revision.
1) Pinus Sylvestris in the title should be in italic type.
2) Clarity and Language:
The manuscript could benefit from a thorough review to improve clarity and language accuracy. Consistency in describing the different ecotypes of Pinus sylvestris L. is crucial for facilitating reader comprehension.
For example, there is an inconsistency between the description in Figure 1, labeled as “6 — Bory Tucholskie NP” and the text at line 72, which refers to the “‘Rychtalska pine’ ecotype (6)”. This discrepancy should be rectified for coherence and accuracy.
3) Readability and Consistency of Tables and Figures:
--- Could the authors explain the meaning of the label “133” for the potential plant community in Table 2?
--- For greater conciseness in Table 1, consider replacing “m2 × 10,000” with “ha”.
--- In line with the logical flow of the manuscript, Table 2 should explicitly indicate “Tree Slenderness Coefficient” values.
--- For statistical rigor, mean values in tables should include either standard deviations or confidence intervals.
4) Methodological Rigor:
--- The differences in levels between tree slenderness coefficient and plant community rankings are not equivalent. Using a summation approach to substantiate these points may lack methodologically rigor.
--- The authors should specify the statistical methods employed for correlation analysis to support the methodological rigor of the study. Spearman correlation analysis? Pearman correlation analysis? Or non-linear relationship?
5) NDVI has a complex relation to the ecological factors. It suggests to study the correlation analysis of NDVI and soil types.
6) There is no correlation between TSC and plant community, between TSC and NDVI. Could the authors explain it?
No
Author Response
Response to the Reviewers to the manuscript forests-2599299: “Prospects for the Preservation of the Main Pinus sylvestris L. Ecotypes in Poland in the Context of the Habitat Conditions of their Occurrence”.
We would like to thank all reviewers for their insightful and substantively valuable comments that allowed us to make our manuscript better. At the same time, we attach below our responses to the Reviewers' comments.
Reviewer 2 (R2)
he manuscript entitled “Prospects for the Preservation of the Main Pinus sylvestris L. Ecotypes in Poland in the Context of the Habitat Conditions of their Occurrence” describes a new approach to assess the sustainability of Pinus sylvestris L. stands in their natural habitat conditions through the analysis of 7 major selected ecotypes and their corresponding environmental conditions. This work could provide valuable information for recognizing the developmental prospects of major Pinus sylvestris L. ecotypes in Poland. I recommend to a minor revision.
(R2): 1) Pinus Sylvestris in the title should be in italic type.
Authors: The change has been made
(R2): 2) Clarity and Language:
The manuscript could benefit from a thorough review to improve clarity and language accuracy. Consistency in describing the different ecotypes of Pinus sylvestris L. is crucial for facilitating reader comprehension.
For example, there is an inconsistency between the description in Figure 1, labeled as “6 — Bory Tucholskie NP” and the text at line 72, which refers to the “‘Rychtalska pine’ ecotype (6)”. This discrepancy should be rectified for coherence and accuracy.
Authors: We would like to thank the reviewer for spotting the error in line 72. The numbering of ecotypes in the text has been changed
(R2): 3) Readability and Consistency of Tables and Figures:
--- Could the authors explain the meaning of the label “133” for the potential plant community in Table 2?
Authors: We would like to thank the reviewer for spotting the error in Table 2. We have replaced “133” with “Oligotrophic Quercus robur-Pinus sylvestris forest (Querco roboris-Pinetum)”
(R2): --- For greater conciseness in Table 1, consider replacing “m2 × 10,000” with “ha”.
Authors: We have replaced “m2 × 10,000” with “ha”.
(R2): --- In line with the logical flow of the manuscript, Table 2 should explicitly indicate “Tree Slenderness Coefficient” values.
Authors: Tree Slenderness Coefficient (TSC) was indicated in title of Table 2
(R2): --- For statistical rigor, mean values in tables should include either standard deviations or confidence intervals.
Authors: Standard deviations added. In order to ensure data transparency, the maximum values, not used in the interpretation of the results, have been removed from the table in Table 3 and standard deviations have been inserted in their place.
R2): --- 4) Methodological Rigor:
R2): --- The differences in levels between tree slenderness coefficient and plant community rankings are not equivalent. Using a summation approach to substantiate these points may lack methodologically rigor.
Authors: The reviewer is right that the adopted levels between tree slenderness coefficient and plant community rankings may be questionable. Therefore, we have included this issue in the Discussion section with a broader explanation.
R2): --- The authors should specify the statistical methods employed for correlation analysis to support the methodological rigor of the study. Spearman correlation analysis? Pearman correlation analysis? Or non-linear relationship?
Authors: We used Pearson's correlation coefficient. We've added this information to the Methods section
R2): --- 5) NDVI has a complex relation to the ecological factors. It suggests to study the correlation analysis of NDVI and soil types.
Authors: The reviewer is absolutely right that the NDVI value is the result of a complex set of ecological factors, which is why we tried to summarize it in Figure 2. The figure shows a certain relationship between soils and NDVI, indicating lower values for podzolic soils and higher for rusty soils. This would confirm the higher trophicity of rusty soils than podzolic ones. We will try to prove these relationships with a larger sample of data in a separate publication.
6) There is no correlation between TSC and plant community, between TSC and NDVI. Could the authors explain it?
Authors: We answered the Reviewer's question in the Discussion section by adding a paragraph dedicated to it.
Reviewer 3 Report
Introduction
Line 37 – the Scots pine
Pine ecotype – it is not clear what the term ecotype means. I suggest Introduction should be expanded with the paragraph about classification of pine ecotypes.
Lines 38-49 – It is unclear why the local pine ecotype is described in such detail, while the distribution range of Scots pine is much wider than Poland and Latvia. Please, add a few general sentences about the object of research (Pinus sylvestris) with good and relevant references.
Materials and Methods
The set of data analyzed is too poor. The study is based on the analysis of a single index (NDVI) calculated for seven sites, which did not allow for reliable statistical analyses. All data on the sites are from published sources, and this fact severely limits the novelty of the study.
Why don’t you analyzed the connection between NDVI and stand parameters from table 2 (tree height and diameter) or other stand parameters important for the assessing pine populations? What is the potential plant community? Why it is “potential”?
I think, the manuscript cannot be published in FORESTS in the present form.
Results
The materials and methods used did not allow the authors to obtain interesting results. The manuscript looks like a brief summary, not a research article.
All abbreviations in tables should be explained (Av.H, Av.DBH and others)
Discussion
There is no need to discuss the dependence of pine on soil type, nor the general patterns of tree species changes. These are obvious things.
A more depth analysis of the relationship between NDVI and stand parameters could have yielded more interesting results, but the authors limited themselves to simplified descriptive study. In present form, I recommend this manuscript to be submitted to more local journal.
References
There are only 21 references. It is confusing for an object as well-studied as a pine.
minor editing of passive voice and typos
Author Response
Response to the Reviewers to the manuscript forests-2599299: “Prospects for the Preservation of the Main Pinus sylvestris L. Ecotypes in Poland in the Context of the Habitat Conditions of their Occurrence”.
We would like to thank all reviewers for their insightful and substantively valuable comments that allowed us to make our manuscript better. At the same time, we attach below our responses to the Reviewers' comments.
Reviewer 3 (R3)
R3 – Introduction
Line 37 – the Scots pine
Authors: We would like to thank the reviewer for spotting the error
R3 – Pine ecotype – it is not clear what the term ecotype means. I suggest Introduction should be expanded with the paragraph about classification of pine ecotypes.
Authors: The Reviewer's comment in the Introduction section has been taken into account
R3 – Lines 38-49 – It is unclear why the local pine ecotype is described in such detail, while the distribution range of Scots pine is much wider than Poland and Latvia. Please, add a few general sentences about the object of research (Pinus sylvestris) with good and relevant references.
Authors: It was assumed that general information about Pinus sylvestris is known, therefore the focus was on local pine ecotyes, which are the subject of the work, emphasizing their importance in relation to general knowledge about pine. General information about Scots Pine is included in the cited literature under items 1, 2, 3, 4, 5. We do not claim that the range of the pine is limited to Poland and Latvia. Drawing attention to these two countries results from the cited literature (Daszkiewicz, P.; Oleksyn, J. Introduction of the "Riga pine" in the 18th and 19th century France. Rocznik Dendrologiczne 2005, 53, 7–40) emphasizing the importance of pine ecotypes from the described region. We found it worth emphasizing the fact that the Riga pine population has survived to this day and that its future is threatened in its natural conditions. This fact is also related to the purpose of the work.
R3 – Materials and Methods
R3 – The set of data analyzed is too poor. The study is based on the analysis of a single index (NDVI) calculated for seven sites, which did not allow for reliable statistical analyses. All data on the sites are from published sources, and this fact severely limits the novelty of the study.
Authors: We wrote in section 2.2 that the choice of NDVI was dictated by the results of our research in which we tested 249 vegetation indices, which were published by MÅ‚ynarczyk, A.; Konatowska, M.; Królewicz, S.; Rutkowski, P.; Piekarczyk, J.; Kowalewski, W. Spectral Indices as a Tool to Assess the Moisture Status of Forest Habitats. Remote Sens. 2022, 14, 4267, https://doi.org/10.3390/rs14174267 [12]. We decided that there was no need to retest the data using all indicators, we just adopted the one that was best.
The data on the sites are from UNPUBLISHED SOURCES and no one has yet compiled and analyzed the data we provided. No one has analyzed NDVI for the studied stands, so no one has previously been able to analyze NDVI in relation to the remaining data.
R3 – Why don’t you analyzed the connection between NDVI and stand parameters from table 2 (tree height and diameter) or other stand parameters important for the assessing pine populations? What is the potential plant community? Why it is “potential”?
Authors: The reviewer was right that it was possible to calculate an additional correlation between the height and the average of the trees, so we calculated them and presented the results in Table 5. The relationship between the height of the trees and other features describing the studied stands is also illustrated in Fig. 2
The potential plant community is the expected plant association in the absence of human intervention. In natural forest ecosystems, potential vegetation is equal to actual vegetation. In disturbed forest ecosystems (e.g. in pine monocultures planted on land used for agriculture in the past), the current plant community may differ from what it would be if the natural natural system had not been disturbed. We try to compare current and potential vegetation by assessing the species composition of all plant species occurring in the described area, in particular species serving as phytosociological identifiers. In the case of the studied stands, phytosociological identifiers potentially indicate plant communities other than natural pine forests. In the described cases this does not have to be the result of disturbances, but e.g. of climate change, although the role of humans in the presence of pine in the studied stands cannot be ruled out, despite the advanced age of the studied stands.
R3 – Results
R3 – The materials and methods used did not allow the authors to obtain interesting results. The manuscript looks like a brief summary, not a research article.
Authors: Scientific articles should be synthetic, which is why scientific journals often impose word or page limits.
R3 – All abbreviations in tables should be explained (Av.H, Av.DBH and others)
Authors: The reviewer's comment is right. Missing explanations of abbreviations have been added
Discussion
R3 – There is no need to discuss the dependence of pine on soil type, nor the general patterns of tree species changes. These are obvious things.
Authors: It is known that pine is a very plastic species and can grow on very different soils, from sandy to clay and from dry to swampy, but the discussion on changes in the species composition of the tree stand was not intended to demonstrate the relationship between pine and soil conditions, but to show that on soils considered suitable for pine develop much richer plant communities than pine forests. And this is no longer an obvious thing. If this were an obvious thing, we would not have such a large share of pine monocultures in potential deciduous forest habitats. Additionally, the criteria for determining potential vegetation on sandy soils if deciduous stands have been replaced by pine monocultures are very poorly described in the literature, which also affects the species composition of the herbaceous and moss layer. Therefore, we believe that the example described in the discussion is important both from the point of view of forest management and from the point of view of identifying plant communities. And since the geographical range of the pine is extensive, the described problem is not of a local nature, but may be of wider interest to Forests journal readers..
R3 – A more depth analysis of the relationship between NDVI and stand parameters could have yielded more interesting results, but the authors limited themselves to simplified descriptive study. In present form, I recommend this manuscript to be submitted to more local journal.
Authors: As written above, the problem is not local and the presented results concern not only forestry, but also phytosociology and the management and protection of forest ecosystems. Forest management is carried out not only in Poland and pine stands grow not only in Poland. Moreover, the described problem does not only concern Scots pine. Therefore, readers in countries where other pine species are grown may also be interested in the topic. Pine monocultures were introduced, among others, also in South American countries, where pine replaced parts of the Amazon jungle. Therefore, it cannot be said that the work is local.
R3 – References
R3 – There are only 21 references. It is confusing for an object as well-studied as a pine.
Scots pine is well-studied species and there is a lot of paper on this pine, by we have cited only the papers directly related to the topic of the study.
Authors: The fact that only 21 works were cited is due to the fact that there are few works on the topic related to the purpose of the research directly. However, the scope of the cited literature has been expanded, but it does not affect either the results or the conclusions.
Round 2
Reviewer 1 Report
The authors improved the ms on a great way and for me, is ready to be published.